# Robust lasing modes in coupled colloidal quantum dot microdisk pairs using a non-Hermitian exceptional point

Evan Lafalce [1], Qingji Zeng[1], Chun Hao Lin[2], Marcus J. Smith[2,3], Sidney T. Malak[2], Jaehan Jung[2,4], Young Jun Yoon[2], Zhiqun Lin [2], Vladimir V. Tsukruk[2] & Z. Valy Vardeny[1]

Evanescently coupled pairs of microdisk lasers have emerged as a useful platform for studying the non-Hermitian physics of exceptional points. It remains an open question how scalable and versatile such phenomena can be when carried over to other designs. Here we have studied the effect of gain/loss modulation in an evanescently coupled pair of microdisk optical resonators fabricated from solution-processed colloidal quantum dots. The emission spectra of these structures are sensitive to small imperfections, which cause frequency-splitting of the whispering gallery modes. Despite this inherent disorder, we found that when spatially modulating the optical pump to vary the gain differential between the coupled microdisks, the coupling drives the split parasitic intra-cavity modes into coalescence at an exceptional point of the resulting three-mode system. This unusual behavior is rationalized via a Hamiltonian that incorporates the intra-cavity coupling as well as the anisotropic inter-cavity coupling between modes in the microdisk pair.

[1] Department of Physics & Astronomy, University of Utah, Salt Lake City, UT 84112, USA. [2] School of Materials Science and Engineering, Georgia Institute of Technology, Atlanta, GA 30332, USA. [3] Aerospace Systems Directorate, Air Force Research Laboratory, Wright-Patterson Air Force Base, Ohio 45433, USA. [4] Department of Materials Science and Engineering, Hongik University, Sejong 30016, South Korea. These authors contributed equally: Evan Lafalce, Qingji Zeng. Correspondence and requests for materials should be addressed to Z.V.V. (email: val@physics.utah.edu)

Non-Hermitian Hamiltonians are used to describe open quantum systems that exchange energy with their environment[1–4]. In contrast to their closed, Hermitian counter-parts, the family of eigenvalues may be complex, with the imaginary part describing the growth or decay of the amplitudes of the associated eigenfunctions. A surge in interest in non-Hermitian formalisms and experimental model systems has been fueled by the unusual behavior associated with exceptional points (EP), which are non-Hermitian analogs of quantum degeneracies, where both the real and imaginary eigenvalues coalesce[2,3]. EPs occur at points in the complex parameter space of eigenvalues that is controlled by the strength of the imaginary potential and coupling between two or more modes. Not limited to quantum systems, these concepts have been largely explored in classical optical systems, including those based on parity-time (PT) symmetry[5,6], that include microwave and optical microcavities[7–10], waveguides[11,12], and photonic crystals[13,14]. Several counter-intuitive and novel optical phenomena have been realized in these studies, including reversed pump-dependence in lasers[14–17], unidirectional light propagation[18,19] and topologically-driven state exchange[8–10,20]. Circular microcavity lasers that support whispering-gallery modes (WGMs) have been an exceptionally fruitful platform in which to study and utilize the physics of EPs. In particular, the demonstration of selective mode-filtering in PT-symmetric coupled cavities[21] to enforce single-mode operation in microlasers[22,23] highlighted the ability to control the spectral properties of multimode lasers. Additionally, EPs have been used to implement high degrees of chirality and directionality to WGMs[24,25] which has enabled the realization of lasers producing beams with orbital-angular momentum[26], and the elimination of spatial hole burning[27]; all on a micro-chip scale.

These initial observations have impressively demonstrated the power of utilizing EPs to control the behavior of the related non-Hermitian systems. However, these implementations have so far relied on precisely designed and well-controlled material systems to achieve the desired performance. The sensitivity of the EP to the implemented values of index of refraction, optical gain and loss, and the coupling between modes requires all of these parameters to be simultaneously maintained, and only limited attention has been given to the robustness of the EP in the presence of disorder such as thermal modulation or structural imperfection. Being elementary in nature, most applications have involved only two interacting modes. Expanding the flexibility of these materials and architectural requirements is a necessary step in order to obtain more general platforms for utilizing the phase-change-behavior of non-Hermitian optics.

It is desirable to develop more flexible materials systems for photonics, particularly for micro-photonics and on-chip photonics. These include the solution based, chemically tunable materials such as π-conjugated polymers, hybrid perovskites and colloidal quantum dots (CQDs). In particular, CQDs are nano-sized semiconductor particles that possess large optical cross-sections such as stimulated emission and absorption. The emission of CQDs can be tuned to cover a wide spectral range from ultra-violet to infrared regions by simply changing their size or composition, providing flexibility to the design of advanced photonic systems that require lasing emission at specific wave-length[28]. These properties make CQDs a promising photonic component that need be studied and developed for future lasing applications. In addition, the ease-of-fabrication can be utilized to create micro-cavities made of solution-processable CQDs by using standard lithographic techniques and a wide variety of substrates[29]. This shows processing advantages over more commercially mature technologies based on inorganic semiconductor alloys and multiple quantum wells that are grown via high-vacuum film processing and strict requirements of lattice-matching. It is an interesting question as to whether non-Hermitian photonics may be implemented on a less exacting platform such as the CQDs.

In the present work we investigate WGMs from microdisk cavities formed from robust crosslinked assemblies of core/alloyed-shell CdSe/Cd$_{1-x}$Zn$_x$Se$_{1-y}$S$_y$ CQDs fabricated by a scalable lithographic process. The majority of the microdisks studied here exhibit WGM-splitting associated with local defects on the disk circumference, which are a natural result of the high-throughput, flexible design approach. Despite this, we demonstrate here localization/delocalization phase-change behavior in near-field-coupled pairs of microdisks. A study of the emission spectra from microdisk pairs reveals that actively pumped interacting microlasers produce spectra more robust against the appearance of mode-splitting. By spatial variation of gain and loss in the coupled microcavity pairs, we show that the removal of the localized parasitic modes originates from their coalescence at an EP of the non-Hermitian multi-mode system. In particular, we show that the intra-cavity mode splitting can be driven into coalescence through inter-cavity coupling in the presence of gain/loss variation. This phenomenon may potentially be used to circumvent some of the inherent obstacles in using solution-based semiconductor materials for lasing and improve the quality of laser emission by reducing the number of parasitic modes in the output.

## Results

**Properties of CQD microdisks.** To fabricate CQD microdisk lasers, we have adopted a facile one-pot method to synthesize high quality compositional gradient core/alloyed-shell CQDs with an average size of 7.5 ± 0.8 nm (see Methods for detailed synthesis procedure). The gradient shell helps to significantly suppress the Auger recombination, which has been the main obstacle to achieve lasing in CQD films in the past. We then utilize a hybrid top-down and bottom-up approach that we previously developed to fabricate the microdisk laser arrays[29] (details described in Methods). In brief, this approach combines standard photo-lithography that provides up-scalability and precise control of spatial distribution to these CQD assemblies, as well as layer-by-layer deposition to integrate tiny CQDs into photoresist-templated trenches along with the crosslinking of ligands to enhance the mechanical integrity (Supplementary Fig. 1). Figure 1a demonstrates a transmission electron microscopy (TEM) image of closely packed CQDs and Fig. 1b shows a scanning electron microscopy (SEM) image of the microdisk structure.

Figure 1c shows the spectral evolution of the laser emission from an exemplary microdisk with increasing pump intensity. At low intensity, only a broad photoluminescence band (FWHM~25 nm) is observed, displaying a peak near 630 nm. Above the threshold intensity of 29 μJ/cm$^2$, narrow laser modes emerge from the PL spectrum (Supplementary Fig. 2). The free-spectral range is $\Delta\lambda \approx 2.6$ nm and well described by the formula for WGM's, namely $\Delta\lambda = \lambda^2/n_{eff}(\pi D)$ where $n_{eff}$ is the effective mode index and $D$ is the disk diameter ($D = 25 \pm 1$ μm). By converting the spectral units to μm$^{-1}$ and taking the Fourier transform, we extract a series of harmonics of the optical path length $n_{eff}(\pi D)$ (inset), giving the effective index $n_{eff} = 1.85 \pm 0.05$ which is in agreement with the CdSe/Cd$_{1-x}$Zn$_x$Se$_{1-y}$S$_y$ CQD film refractive index ($n = 1.86 \pm 0.05$) determined by ellipsometry[30,31]. In Fig. 1d, we show the angular dependence of the laser emission in the plane of the microdisk, which is isotropic, signaling a homogenous intensity distribution along the microdisk circumference. Additionally, the fluorescent image of the disk under lasing condition shows that the field amplitude is predominantly located along the perimeter.

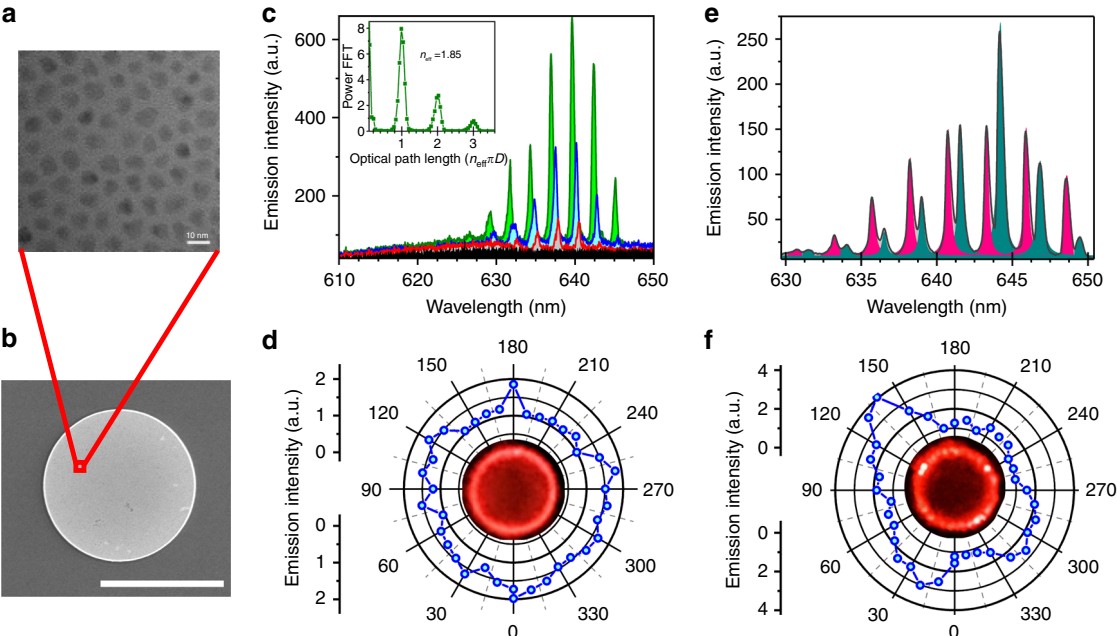

**Fig. 1** Emission characteristics of isolated microdisk resonators. **a** Transmission electron microscopy (TEM) micrograph of oleic acid-capped CdSe/Cd$_{1-x}$Zn$_x$Se$_{1-y}$S$_y$ quantum dots. **b** Scanning electron microscopy (SEM) image of a microdisk (scale bar is 20 μm). **c** Laser emission spectra from a microdisk pumped at various intensities: 16 μJ/cm$^2$ (black), 29 μJ/cm$^2$ (red), 66 μJ/cm$^2$ (blue), 116 μJ/cm$^2$ (green). The inset shows the FFT of the emission spectrum vs. the optical path length $n_{eff}\pi D$, where $n_{eff} = 1.85$ is the effective mode index and $D = 25$ μm. **d** Collection-angle dependence of the emission in the plane of the microdisk in **b** with fluorescent image at the center. **e** Laser emission spectrum from a different microdisk that contains a defect, pumped at 116 μJ/cm$^2$. The different shaded regions are Lorentzian fits to the two laser modes series. **f** Same as in **d** but for the spectrum shown in **e**

In contrast, in Fig. 1e we present the emission spectra from another microdisk of the same size. In this case we are able to resolve two sets of modes, each having $\Delta\lambda = 2.6$ nm. The additional set of modes is split-off from the other WGM modes by ~0.8 nm. The linewidths of all modes are comparable (~0.5 nm) and the splitting does not vary much with increasing pump fluence. In the fluorescent image of Fig. 1f, it is seen that a few particularly bright spots have now appeared on the microdisk circumference.

Microscopic characterization reveals that these bright spots are due to fabrication defects (Supplementary Fig. 3). These defects then act as asymmetric scattering centers that increase out-coupling in the vertical direction giving them their bright appearance. In addition, we see that the defects strongly disrupt the emission isotropy, leading to strong directionality of the emission diametrically across from the defects. Atomic force microscopy has been used to investigate the formation of possible defects during fabrication (Supplementary Fig. 4). There are generally two types of defects in our microdisks that can induce the obtained mode-splitting. The first type is the small CQD aggregates that forms near the disk circumference. These aggregates may originate from the imperfections that exist in CQD solutions, which act as seeds to affect the drying process. The second type of defect is the difference in circumference height due to inhomogeneous drying. Both type of defects may act as localized perturbations that can introduce asymmetric back-scattering between clockwise (CW) and counter-clockwise (CCW) WGM propagation, lifting the degeneracy between them. This degeneracy lifting has been widely discussed in the context of nanoparticle sensing, where the mode-splitting is employed to register the detection event[32].

**Spectral properties of coupled microdisks.** Considering the various phenomena that have been displayed in the non-

Hermitian coupled WGM resonators, it is interesting to consider to what degree these interactions may be preserved in the presence of disorder. This information is imperative if such effects are to be eventually employed in a broad range of photonic applications. We therefore took to the investigation of coupled-pairs of our CQD micro-resonators fabricated from a facile, scalable process. The spacing between the two microdisks was 396 ± 20 nm apart (Supplementary Fig. 5), which is less than the emission wavelength and should provide adequate evanescent coupling between the WGM of the two microdisks. The coupling between microdisks was further verified using finite difference time domain (FDTD) simulations (Supplementary Figs. 6, 7). In Fig. 2a–c, we show the emission spectra obtained when the pair is placed at the center of the beam spot, so that both individual microdisks are pumped evenly (Fig. 2b), and when the left or right disks are pumped exclusively (Fig. 2a, c). This configuration resembles the PT-symmetric laser system that was theoretically proposed[21] and experimentally demonstrated to achieve selective mode-filtering[22]. In all three cases we see a distinct laser emission spectrum, and we particularly note the appearance of mode-splitting in Fig. 2c when only the right disk is pumped. It is therefore surprising that the emission spectrum of the evenly-pumped pair shows no trace of the mode-splitting in the same spectral region (Fig. 2b). This is in stark contrast to the majority of single, isolated microdisks whose spectra contain parasitic mode-splitting. These results suggest that the coupling between the microdisks in the pair is helpful in reestablishing spectral purity in the laser emission spectrum.

In Fig. 2d–f we demonstrate that this phenomenon is robust. It is observed in numerous samples in which at least one of the microdisks exhibits mode-splitting during the asymmetric pumping condition. To facilitate comparison over a large number of coupled-microdisks, we define an empirical parameter, $\phi^\lambda$, to quantify the splitting between individual broken-degeneracy

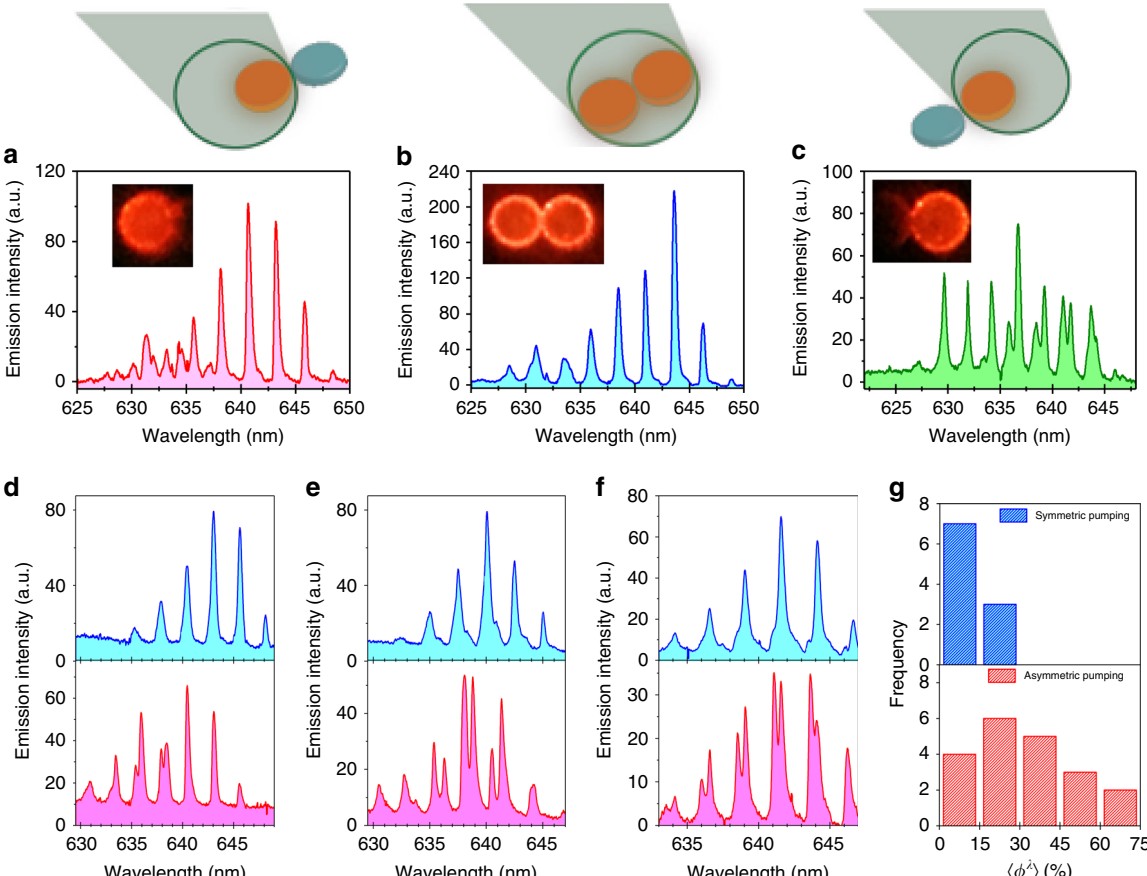

**Fig. 2** Emission characteristics of coupled pairs of microdisk resonators. **a–c** Laser emission spectra from a coupled microdisk pair, where the pair is placed at different locations in the pump beam spot, such that only the left or right microdisk is pumped (**a** and **c**), or the pair is pumped evenly, (**b**), as illustrated schematically above each plot. The insets show the fluorescent image from the microdisks in each case. **d–f** Laser emission spectra from three different microdisk pairs pumped at 116 μJ/cm². In each figure the spectrum from the evenly-pumped pair (blue) is compared to a spectrum from the asymmetric pumping scheme (red). The evenly-pumped pair consistently shows a reduction in mode-splitting compared to the asymmetric case. **g** Frequency distribution of the spectrally averaged modal splitting parameter, $\langle \phi^\lambda \rangle$, (defined in Eq. 1) for the symmetric (blue) and asymmetric pumping schemes (red)

WGM mode-pair,

$$\phi^\lambda = \frac{2|\lambda_1 - \lambda_2|}{\Delta \lambda} \qquad (1)$$

where $\lambda_1$ and $\lambda_2$ are the peak wavelengths, and the wavelength splitting is normalized by half the free-spectral range $\Delta \lambda$. We then take the average $\langle \phi^\lambda \rangle$ of all mode-pairs in the spectrum (see Supplementary Fig. 8). Figure 2g shows the frequency distribution of $\langle \phi^\lambda \rangle$ obtained from both the symmetric and asymmetric pumping configurations. Indeed, the wide variability of the emission from asymmetrically pumped pairs, which echoes the variation in the emission spectrum of isolated microdisks (Supplementary Fig. 9), collapses to a narrower distribution with significantly reduced parasitic mode-splitting when the pair is pumped evenly. The results show that the coupling between microdisks in the presence of symmetric gain consistently corrects the sample variability that naturally results from a scalable, high-throughput fabrication procedure for micro-photonic elements such as these. As we show below, this is possible due to the presence of an EP in the Hamiltonian of the system that provides a channel through which intra-cavity mode splitting can be modulated through inter-cavity coupling.

**Mode coalescence revealed through spatial gain variation.** A detailed look at this behavior is provided by an examination of

the laser emission modes under the application of spatial gain variation (Fig. 3). In these measurements, the microdisk pair is swept through the pump beam spot incrementally. The offset between the pump spot and the pair center, which we call $\Delta D_p$, then becomes a proxy for the gain/loss variation, $\Delta g_{AB} = g_B - g_A$ between the two microdisks, where we refer to the microdisk that shows parasitic splitting as microdisk A with gain, $g_A$ and its neighbor microdisk B with gain, $g_B$. As seen in Fig. 3a, when $\Delta D_p$ is large and negative, only microdisk A is pumped, and clear mode-splitting can be observed in the laser emission spectrum. Then as the pair of coupled microdisks is moved towards the center of the excitation beam spot and gain is added to microdisk B, we find that the localized intra-cavity modes gradually move closer in frequency and eventually coalesce near but not quite at the center. This is very surprising, as these modes are localized in microdisk A and the gain/loss differential does not change between them. Here, the application of gain to microdisk B modulates the splitting between the parasitic defect modes of microdisk A through the evanescent coupling between the two microdisks.

**Eigenvalues of the three-mode non-Hermitian Hamiltonian.** To understand this behavior in more detail, we consider a solution to the eigenvalue problem described by the following

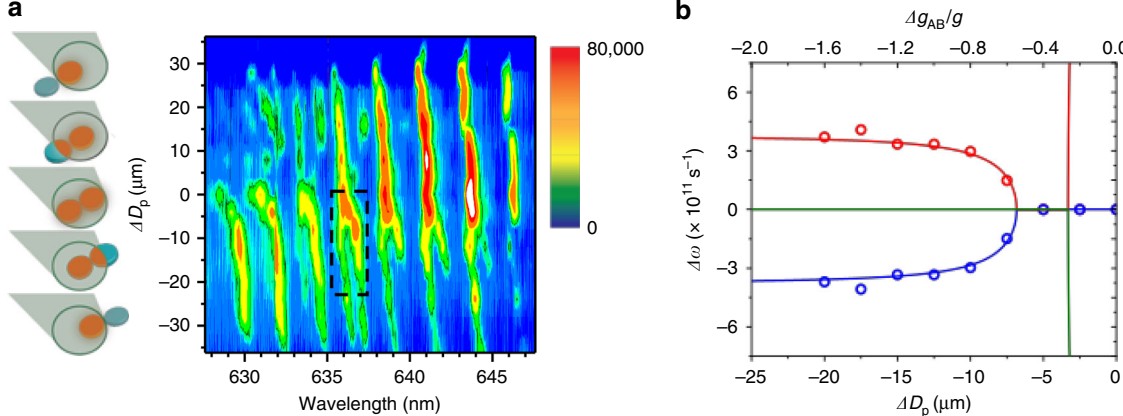

**Fig. 3** Laser emission behavior of a microdisk pair under spatial gain variation **a** False-color contour plot of the emission intensity vs. wavelength and relative distance, $\Delta D_p$, between the center of the pair to the center of the beam spot, as the pair is shifted through the pump beam spot, which is schematically illustrated on the left. As the microdisk pair nears the center, the split modes merge as clearly seen for several mode pairs in the range of 635–645nm. **b** Mode splitting in the defected microdisk vs. $\Delta D_p$ as obtained from the dashed-boxed region in **a**. The solid lines are eigenvalue dynamics as a function of the gain differential $\Delta g_{AB}$ between the coupled microdisks in a three-mode Hamiltonian that shows coalescence of the intra-cavity modes in the defected microdisk due to the gain variation between the coupled microdisks. The model parameters are $\kappa = 3.8 \times 10^{11}\,s^{-1}$, $\gamma_{23} = 1.2 \times 10^{12}\,s^{-1}$ and $\gamma_{13} = 0\,s^{-1}$

Hamiltonian:

$$\hat{H} = \begin{pmatrix} \omega'_1 + ig_A & \kappa & \gamma_{13} \\ \kappa & \omega'_2 + ig_A & \gamma_{23} \\ \gamma_{13} & \gamma_{23} & \omega'_3 + ig_B \end{pmatrix} \quad (2)$$

where $\omega'_{i=1,2,3}$ are the real parts of the eigenfrequencies in the absence of coupling and $g_{i=1,2,3}$ are the imaginary parts, where we set $g_1 = g_2 = g_A$, and similarly we set $g_3 = g_B$ for the mode 3 in microdisk B. $\kappa$ is used to represent the intra-disk coupling between CW and CCW modes in microdisk A (modes 1 and 2), whereas $\gamma_{i3}$ describes inter-disk coupling of $i$th mode of microdisk A to the mode 3 of microdisk B (see Fig. 4a). This is an extension of the $2 \times 2$ Hamiltonians often considered in non-Hermitian systems, where an interaction exists between one mode from each resonator[22]. In our case the defect-induced mode-splitting produces a scenario in which three modes interact, and the resulting Hamiltonian is $3 \times 3$. In order to simulate the portion of the spatial gain variation experiment in which $\Delta D_p$ increases from $-25\,\mu m$ to zero we keep $g_A$ constant, $g_A = g$ (microdisk A is entirely covered by the pump beam) while sweeping $g_B$ from $-g$ to $g$ (microdisk B goes from unpumped to completely pumped). Thus, $\Delta g_{AB}$ ranges from $-2\,g$ to zero.

Considering the boxed region in Fig. 3a, the experimentally observed phenomenon of intra-cavity mode coalescence can be satisfactorily replicated as demonstrated in Fig. 3b. The simulations show a typical branch root behavior of an EP that is the origin of the observed coalescence. By setting $\omega'_1 = \omega'_2 = \omega'_3 = \omega_0$, where $\omega_0$ is the average frequency of all three modes, the splitting at large $\Delta g_{AB}$ is determined by the intra-cavity mode coupling $\kappa$. The values of $\omega_0$, $g$, and $\kappa$ are then taken from experiment[30], leaving only $\gamma_{23}$ and $\gamma_{13}$ as free parameters. As discussed below, the key to the observation of the coalescence of intracavity modes is the existence of a large coupling anisotropy, $\gamma_{23} \gg \gamma_{13}$, with an exact EP obtained in the limit $\gamma_{13} \rightarrow 0$. Similar agreement is obtained between the experimental results and the model for each of the modes for which the coalescence is observed by adjusting the coupling term $\gamma_{23}$ (Supplementary Fig. 10).

We have analyzed the Hamiltonian in detail to extract the characteristic behavior of the eigenvalues (Supplementary Figs. 11–13). Shown in Fig. 4b, it is seen that when the absolute value of the gain differential $|\Delta g_{AB}|$ is large, the modes are localized in respective disks and the splitting is determined by the intra-cavity coupling factor of microdisk A taking a value of $2\kappa$. As $|\Delta g_{AB}|$ decreases, the splitting is reduced, showing the square root dependence on $\Delta g_{AB}$ that is the signature of EP behavior[2,3]. At the particular value $\Delta g_{AB} = [\kappa + (\gamma_{23})^2/2\kappa]$ the eigenvalue splitting collapses to $\gamma_{13}/2$. Thus, when $\gamma_{13} = 0$, the eigenvalues coalesce and the point corresponds to an EP, denoted as $EP_1$. In accordance, the imaginary parts of modes 1 and 2 bifurcate and mode 2 becomes more lossy. Note the remarkable implications of this result, as also shown in the experiment, that the observed splitting of intra-cavity modes of microdisk A depends on the gain applied to microdisk B. As $|\Delta g_{AB}|$ decreases further, another transition occurs when $\Delta g_{AB} = 2\gamma_{23}$, denoted as $EP_2$. At this point the real parts of the eigenfrequencies are repelled by the inter-cavity interaction and the frequency splitting becomes $2\gamma_{23}$. Meanwhile the imaginary parts of eigenvalues 2 and 3 coalesce at this point. This transition is similar to the typical transition between PT-symmetric and PT-broken phases studied in other systems[19,22]. From our model we conclude that the coalescence of intracavity-modes and resulting purification of the laser spectrum results from a natural asymmetry between $\gamma_{23}$ and $\gamma_{13}$. Such an asymmetry in coupling coefficients may arise from the 90° phase shift expected[32,33] between broken-degeneracy modes 1 and 2, leading to maximal anti-node to anti-node overlap between modes 2 and 3 and minimal overlap between anti-nodes of modes 1 and 3.

## Discussion

Because both eigenvalues and eigenvectors become degenerate at an EP, the WGM modes also become unidirectional[24–27]. While free-space coupling employed in our system precludes determination of the directionality, we note that EPs resulting from the interaction between three modes were discussed theoretically by Demange and Graefe[34]. They identified two types of EPs in this case: a three-mode third root branch point and a square root branch point between two of the three modes, that shows similar

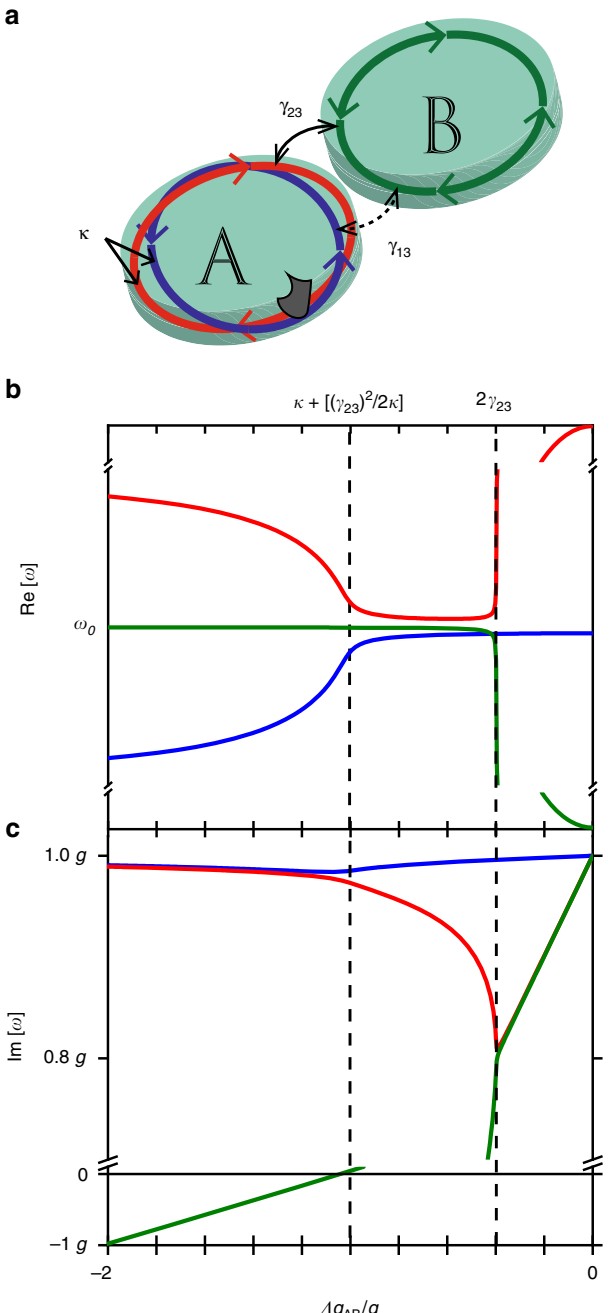

**Fig. 4** Calculated eigenvalue dynamics of the three-mode system. **a** Schematic diagram of the modes considered and parameters governing their interaction. For microdisk A, a defect on the circumference (shown as the grey area) induces splitting between modes due to their mutual asymmetric backscattering determined by the parameter $\kappa$. These two modes can then couple to the degenerate modes in microdisk B through the inter-disk coupling parameters, $\gamma_{13}$ and $\gamma_{23}$. We consider the case for which $\gamma_{13} \ll \kappa \ll \gamma_{23}$. **b**, **c** Real and Imaginary parts of the eigenvalues of the three-mode Hamiltonian vs. the normalized gain differential, $\Delta g_{AB}/g$ between microdisks A and B. The dashed lines indicate the locations of the exceptional points $EP_1$ and $EP_2$. The limiting values of the eigenvalue splitting are $2\kappa$ at $\Delta g_{AB} = -2\,g$ and $2\gamma_{23}$ at $\Delta g_{AB} = 0$, while the minimum splitting between EP1 and EP2 is $\gamma_{13}/2$

behavior to EPs involving only two modes. Since the $EP_1$ described in this work corresponds to the latter case, we expect the directional properties to behave in similar manner to previous

two mode systems[24–27], namely, that modes 1 and 2 will become unidirectional and power flow will be chiral at $EP_1$.

It is worth pointing out the difference between our results and those of Hodaei et al.[22] who used a similar system. In particular, there it was shown that the suppression of multi-mode emission can be achieved by engineering the PT-symmetric system in the asymmetric pumping scheme. In our samples the disorder-induced intra-disk coupling seems to circumvent such mode-suppression. In fact, we have observed, in cases where at least one disk is free of defects, significant suppression of multi-modal emission (Supplementary Fig. 14). Although we have emphasized the spectral mode control through the EP in disordered cavities with naturally occurring defects, we have also observed similar effects by incoporation of synthetic notch defects into the cavity design and photolithographic fabrication process. With more direct control of the size of the defect, the degree of mode-splitting can be tuned and strong directionality can be imparted (Supplementary Fig. 15). In notched microdisk pairs where the mode splitting has been removed due to implementation of a large notch, mode-splitting is absent in the asymmetric pumping scheme but is observed when the pair is symmetrically pumped (Supplementary Fig. 16). A detailed description of localization/delocalization behavior through the interplay between intra-disk and inter-disk coupling in multimode resonators is the goal of future investigations using synthetic notch implants and holds great promise for harnessing more of the exotic physics of EPs.

The higher-dimensionality of parameter space involved in our coupled-resonator system may be useful for nanoparticle sensing applications. It has been demonstrated that a WGM resonator interacting with two judiciously oriented particles may prepare the system at an EP, providing enhanced sensitivity for detection of a third, target particle[33,35,36]. Since in our case, the coalescence of modes depends on the gain differential with a third-party, this phenomenon may provide an alternative module capable of using an EP for particle detection with increased sensitivity. The coupled-resonator system may be more appealing from an application design point of view due to its simpler implementation and because it does not add additional extrinsic loss. Alternatively, it should be possible to utilize the EP behavior to remove unwanted parasitic modes of individual microcavities in coupled resonators. The applied gain differential could be used as a knob to fine tune the quality of the emission spectrum. Implementation of EPs for this purpose can be particularly welcome benefit for semiconducting photonic materials such as CQD's and $\pi$-conjugated polymers, helping to mitigate the trade-off of disorder and instability that comes with their processing advantages.

## Methods

**Chemicals and materials**. Cadmium oxide, tri-n-octylphosphine (TOP, 90%), 1,7 diaminoheptane (DIAH, 98%), ethyl lactate, zinc acetate, dodecanethiol and selenium powder were obtained from Sigma Aldrich. 1-octadecene (ODE, 90%), butylamine (BA, 98%), oleic acid (OA, 97%), and sulfur were obtained from TCI. Toluene and heptane were obtained from BDH Chemicals. CYTOP was obtained from AGC Chemicals. NR 71-3000p photoresist was purchased from Futurrex. All chemicals were used as received.

**Synthesis of compositional gradient CdSe/Cd$_{1-x}$Zn$_x$Se$_{1-y}$S$_y$ QD**. CdSe/Cd$_{1-x}$Zn$_x$Se$_{1-y}$S$_y$ core/alloyed-shell QDs feature a CdSe core encased in a Cd$_{1-x}$Zn$_x$Se$_{1-y}$S$_y$ alloyed-shell, where $x$ and $y$ increase from zero at the shell interior to 1 at the shell outer-surface. These QDs were synthesized by modifying reported methods[37,38]. The compositional gradient shell is created by using the reactivity difference between Cd and Zn precursors and that between Se and S precursors. Briefly, 1 mmol of CdO, 2 mmol of Zn(acetate)$_2$, 5 ml of oleic acid, and 15 ml of 1-octadecene (ODE) were placed in a three-neck flask and degassed at 150 °C for 1 h. The reaction was heated to 300 °C under Ar. At the elevated temperature (300 °C), 1 ml of 1 M Se/TOP solution was rapidly injected to initiate nucleation and growth. After 5 min, 0.3 ml dodecanethiol was added. The reaction was kept at 300 °C for 20 min. Then, 1 ml of 2 M S/TOP was added. Then the heating mantle was removed to stop reaction. Ten ml of

hexane was added to the solution once the temperature reached 70 °C. The QD solution was purified by adding acetone as an antisolvent to remove excess ligands. After discarding the supernatant, QDs were dissolved in heptane and stored in a refrigerator.

**Microdisk fabrication**. Fabrication of QD microdisks includes several steps. First, a low refractive index layer of CYTOP ($n = 1.34$) was deposited on the silicon wafer ($n = 3.44$) in order to provide light confinement within the QD cavities. CYTOP solution was spun cast on the substrate with a spin speed of 2500 rpm for 3 min. A subsequent baking at 100 °C for 30 min was performed. This process was repeated two times to achieve a film thickness of 1.5 μm. A short oxygen plasma etch (5 s) was performed to improve the wettability of the CYTOP surface for the deposition of the negative photoresist (NR 71-3000p). Ethyl lactate was added to the negative resist NR71-3000p solution to dilute it to one third of the original concentration provided by the company. The diluted resist was spun cast on the CYTOP/silicon substrate (3000 rpm for 1 min). The cast film was subsequently soft baked at 165 °C for 5 min and exposed to 365 nm with a dosage of 123 mW. The exposed film was then post-baked at 100 °C for 5 min and developed by soaking in RD6 developer for 5 s. After the development, the film was rinsed with water and dried by blowing with air. The QD microdisks were fabricated by spin casting butylamine-capped QD solution (in heptane) of ~3–6 mg/mL at 1000 rpm for 1 min onto the polymer pattern. The cast layer was subsequently immersed in 0.1 M diaminoheptane solution in methanol for 1 min and rinsed with methanol two times while spinning at 3000 rpm for 1 min. The above process was repeated multiple times to achieve the desired thickness. The polymer pattern was subsequently removed by soaking in acetone while sonicating for 3–10 s. The characterization of gain properties of films was previously described[30].

**Atomic force microscopy**. AFM topography and phase images were collected using a Dimension Icon AFM microscope (Bruker) equipped with a Nanoscope V controller in tapping mode according to the usual procedure[39]. Briefly, Mikro-Masch rectangular n-type Si probe chips containing four cantilevers each were used. Of the four, cantilever C having a spring constant of approximately 7 N/m was used for AFM scans. Following cantilever tuning via frequency sweep, AFM scan sizes of 30 μm by 30 μm with a scan rate within 0.3–0.8 Hz, and resolution of 512 or 1024 samples per line were conducted.

**Transmission electron microscopy**. The size of CdSe/Cd$_{1−x}$Zn$_x$Se$_{1−y}$S$_y$ QDs was studied using a high-resolution transmission electron microscope (Tecnai F30). An accelerating voltage of 300 keV was used. TEM samples were prepared by diluting the original QD solution of ~5 mg/mL 50 times. Ten μl of the diluted solution was then drop-cast on the TEM grid and allowed to dry completely.

**Scanning electron microscopy**. SEM characterization was performed on a Hitachi S-3400N SEM with a back-scattering electron detector with an accelerating voltage in the range of 5–10 kV.

**Optical characterization**. The samples were placed on a three-dimensional stage under a home-built confocal micro-photoluminescence set-up. The excitation source was provided by the second harmonic (532 nm) of a solid-state pulsed laser based on Nd:YAG delivering 7 ps pulses at a repetition rate of 200 Hz. The beam was focused through a 40× (NA = 0.65) microscope objective onto the sample using a dichroic mirror (550 nm long-pass). The emission from the sample was collected through the same objective transmitted through the dichroic mirror as well as an additional 550 nm long-pass filter, focused onto a multimode optical fiber and recorded using a 1/2 m spectrometer and CCD array. The spectral revolution of the setup was 0.2 nm. Alternatively, a camera was placed in the emission path to collect PL images of the disks. The beam size was set with an iris to 50 μm in diameter.

**FDTD simulations**. The LUMERICAL FDTD software package was used to simulate mode coupling between microdisks of diameters of $D = 26$ μm. The refractive index was set to $n = 1.9$ and a WGM at 630 nm (in vacuum) was simulated. Coupling rates were extracted by pumping one microdisk and observing the rate of amplitude increase in an unpumped neighboring microdisk separated by various gap spacings in the range of 150–400 nm. For the visualization of mode field profiles, two microdisks, with diameters of $D = 4$ μm, were pumped equally.

## Data availability

The data that support the findings of this study are available from the corresponding author upon request.

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

## Acknowledgements

This work was supported by the Air Force Office of Scientific Research (AFOSR) (MURI FA 9550-14-1-0037). C.H.L., Z.L., and V.V.T. acknowledge supports for synthesis, fabrication and characterization of QD and photonic arrays from the Department of Energy (DOE) Office of Science (DE-SC0002232), Air Force Office for Scientific Research (AFOSR) (FA9550-16-1-0187 and FA9550-17-1-0297), and National Science Foundation (NSF) (NSF-CHE-1506046). M.J.S. would like to acknowledge the Science, Mathematics and Research for Transformation (SMART) scholarship funded by OSD-T&E (Office of Secretary Defense-Test and Evaluation), Defense–Wide/PE0601120D8Z National Defense Education Program (NDEP)/BA-1, Basic Research, SMART Program office Grant Number N00244-09-1-0081.

## Author contributions

Synthetic procedures including particle synthesis and solution-phase ligand exchange were developed and performed by J.J. and Y.J.Y. J.J. also characterized the size distribution of QDs using TEM. Microfabrication of single and coupled microdisk lasers was done by C.H.L. Characterization measurements including optical microscope, AFM and SEM were performed by M.J.S. and S.T.M. Optical characterization measurements were designed and performed by E.L. and Q.Z. E.L. analyzed the data, performed eigenvalue calculations, and wrote the manuscript. Q.Z. performed FDTD simulations. Z. L., V.V.T. and Z.V.V. supervised the project, discussed research plans, results and data, discussed and edited the manuscript.

## Additional information

**Competing interests:** The authors declare no competing interests.

