## [Peer Review File · Nature Communications]

Reviewers' comments:

Reviewer #1 (Remarks to the Author):

"Robust lasing modes in coupled colloidal quantum dot microdisk pairs using a non-Hermitian exceptional point" by E. Lefalce, et al.

This work investigates experimentally, the effect of gain and loss modulation in evanescently coupled microdisk resonators fabricated through solution-processed colloidal quantum dots. Such a coupled cavity system has been intensely explored in recent years given its importance as a fundamental non-Hermitian unit governed by a simple matrix Hamiltonian. One of the most important properties of such a Hamiltonian is the emergence of an exceptional point singularity that is a unique feature of non-Hermitian Hamiltonians. The authors have investigated under different pumping two coupled active microdisc resonators when one resonator exhibits a defect which breaks the degeneracy of its counter-rotating modes. Therefore, three modes (and their different longitudinal variations) interact in this system. The authors explore the effect of non-uniform pumping on the evolution of the radiation peaks. The experimental results suggest that when two discs (one with perturbation and the other without) are equally pumped, a better spectral purity is observed since no mode splitting happens.

Compared to the previous experiments, the present manuscript has two novelties: (i) it is based on colloidal quantum dots, and (ii) it involves both inter- and intra-cavity coupling between modes. The experimental results are clean and in good agreement with coupled mode theory and the manuscript is clearly written.

I recommend publication of this manuscript in Nature Communications pending a few minor suggestions:

- The separation of ~ 396 nm between the two resonators, seems a bit far to create evanescent coupling between whispering gallery modes, considering that such modes have a very small waist. This can also be seen in Fig. 2b which shows no mode splitting. The authors have mentioned the absence of mode splitting in this case. I encourage the authors to somehow quantify this coupling through simulations.
- Relevant to the previous point, it would be nice to show the field profile of the resonance modes of the microdiscs which could be a good way to see if there is any overlap between the fields at two discs separated by ~ 400 nm.
- The choice of the empirical parameter ϕ , as defined in Eq. (1), is a rather unusual definition. If I correctly understand this A_1 and A_2 are the intensity of peaks in the mode doublets while $\lambda_2 - \lambda_1$ is the wavelength splitting. On the other hand, both parameters can be obtained from coupled mode theory since the amplitudes come from the eigenvectors and the splitting comes from the eigenvalues. Therefore, the authors could separately present two parameters A_2/A_1 and $\lambda_2 - \lambda_1$. I see no reason to combine those.
- The label for the y-axis of Figure 3 is a bit confusing because it is frequency expressed in inverse centimeters. They should either present wavenumber or report results in terms of inverse second. Similarly, "kappa" and "gamma" should be reported in inverse second.
- Regarding the presentation, there are potential readers from mathematical physics, therefore it would be nice to better present the experimental platform. I suggest authors bringing the supplementary figure S1 to the main text and perhaps even adding an illustrative schematic of their TEM image.
- It would be fair to mention the following reference which is the first work suggesting (theoretically) the use of coupled-cavity PT laser for selective filtering of modes: Miri, M. A., LiKamWa, P., & Christodoulides, D. N. (2012). Large area single-mode parity-time-

symmetric laser amplifiers. *Optics Letters*, 37(5), 764-766

Reviewer #2 (Remarks to the Author):

The manuscript by Evan Lafalce et al presents the spectrum purification of a defect micro-disk laser by forming exceptional points with another micro-disk laser. While the defect-introduced mode splitting between CW and CCW modes have been investigated, the results presented here are innovative towards exceptional point-related novel control on laser actions. Overall, I found the results are suitable for *Nature Communications* if the following comments and suggestions are well addressed:

(1) In the fabrication of the devices, the synthesis method with multiple chemicals presented in reference [22] was applied. While the fabrication approach is different, a similar coupled cavity structure using InGaAsP was described in [16] where single mode lasing was demonstrated. My question is: why using a very similar configuration, different results with multi-mode lasing were achieved (for example, Figure 2)?

(2) Unidirectional lasing may be expected if either CW or CCW mode is removed from the spectrum. Has this been tested? The author could comment more in-depth on the characteristics of the degenerated mode at the exceptional point.

(3) The manuscript should clearly present how the defect works to split the CW and CCW modes. The defect is not visualized in the SEM pictures. How was the defect designed and fabricated?

(4) The lasing condition is determined by the threshold pumping intensity which is 29 $\mu\text{J}/\text{cm}$ stated in line 112. How was the threshold intensity decided? The light-light curve (the I-V curve) should be attached. This is important because we may not be able to tell the differences between the disordered spectrum of illumination under the lasing condition and the spectrum splitting.

(5) In line 160, more convincing results should be given to the coupling between micro-disks to validate that the evanescent field can provide adequate coupling. For example, how much, with a certain spacing, is the coupling strength between mode 1 and 3, mode 2 and 3, respectively?

(6) The introduction part reads great for presenting the concepts of non-Hermitian system and exceptional points. To make the introduction more complete, I suggest the authors including a few references on exceptional point lasers, such as *Science* 353, 464 (2016); *PNAS* 113, 6845 (2016); *Opt. Lett.* 39, 2423 (2014); and a recent ASAP publication in *ACS Photonics* DOI: 10.1021/acsp Photonics.8b00800.

In conclusion, I think the subject itself is interesting that the spectrum splitting by defects can be turned off through exceptional points formed by the asymmetric coupling with a third mode.

Reviewers' comments:

Reviewer #1 (Remarks to the Author):

“Robust lasing modes in coupled colloidal quantum dot microdisk pairs using a non-Hermitian exceptional point” by E. Lefalce, et al.

This work investigates experimentally, the effect of gain and loss modulation in evanescently coupled microdisk resonators fabricated through solution-processed colloidal quantum dots. Such a coupled cavity system has been intensely explored in recent years given its importance as a fundamental non-Hermitian unit governed by a simple matrix Hamiltonian. One of the most important properties of such a Hamiltonian is the emergence of an exceptional point singularity that is a unique feature of non-Hermitian Hamiltonians. The authors have investigated under different pumping two coupled active microdisc resonators when one resonator exhibits a defect which breaks the degeneracy of its counter-rotating modes. Therefore, three modes (and their different longitudinal variations) interact in this system. The authors explore the effect of non-uniform pumping on the evolution of the radiation peaks. The experimental results suggest that when two discs (one with perturbation and the other without) are equally pumped, a better spectral purity is observed since no mode splitting happens.

Compared to the previous experiments, the present manuscript has two novelties: (i) it is based on colloidal quantum dots, and (ii) it involves both inter- and intra-cavity coupling between modes. The experimental results are clean and in good agreement with coupled mode theory and the manuscript is clearly written.

I recommend publication of this manuscript in Nature Communications pending a few minor suggestions:
- The separation of ~396 nm between the two resonators, seems a bit far to create evanescent coupling between whispering gallery modes, considering that such modes have a very small waist. This can also be seen in Fig. 2b which shows no mode splitting. The authors have mentioned the absence of mode splitting in this case. I encourage the authors to somehow quantify this coupling through simulations.

Response: We performed Finite Difference Time-Domain (FDTD) simulations using the LUMERICAL software package to investigate coupling between WGM modes in our microdisks, as a function of the gap spacing. The extracted coupling rate vs distance is displayed in the figure below. The theoretical coupling value in our experimental range of interest is $\sim 7 \times 10^{10} \text{ s}^{-1}$. We provide additional experimental-based evidence of coupling between microdisks in response to Comment (3) of Reviewer #2 that suggests the coupling is larger than the theoretically derived values.

Figure R1 was added to the Supplementary Information as Supplementary Figure S6. It is referenced in the manuscript text on line 161 on page 7 relative to the original manuscript. A description of the FDTD simulations was added to the methods section.

Figure R1. Coupling rates between microdisks having refractive index $n=1.9$ that were extracted from FDTD simulations for WGM at 630 nm (in vacuum).

- Relevant to the previous point, it would be nice to show the field profile of the resonance modes of the microdisks which could be a good way to see if there is any overlap between the fields at two discs separated by ~ 400 nm.

Response: Our FDTD simulations have also shown overlap of WGM modes in disks separated by 400nm, as shown below in Fig. R2. We note that while we have used a realistic wavelength (630nm) and the experimentally determined index of our system ($n = 1.9$), we have used smaller microdisks in the simulation, namely, $D = 4\mu\text{m}$ instead of the experimental $D = 26\mu\text{m}$, so that the mode profiles are visible. In the experimental system, the reduced curvature would increase the region of modal overlap providing enhanced coupling between the WGMs on two microdisks.

Figure R2 was added to the Supplementary Information as Supplementary Figure S7. It is referenced in the manuscript text on line 161 on page 7 relative to the original manuscript.

Figure R2. (Left panel) Field profiles from the FDTD simulation of two microdisks with diameter $D = 4\mu\text{m}$ separated by a distance of 400nm. (Right panel) Close up view of the region between the microdisks emphasizing the overlap of WGM anti-nodes.

- The choice of the empirical parameter ϕ , as defined in Eq. (1), is a rather unusual definition. If I correctly understand this A_1 and A_2 are the intensity of peaks in the mode doublets while $\lambda_2 - \lambda_1$ is the wavelength splitting. On the other hand, both parameters can be obtained from coupled mode theory since the amplitudes come from the eigenvectors and the splitting comes from the eigenvalues. Therefore, the authors could separately present two parameters A_2/A_1 and $\lambda_2 - \lambda_1$. I see no reason to combine those.

Response: We agree that the combination of amplitude and frequency information is confusing, and have separated the two parameters in the revised manuscript. Because the splitting parameters based on wavelength and amplitude closely track one another, we have presented the statistics on the wavelength based splitting parameter in Fig. 2g, while presenting the statistics based on both amplitude and wavelength in the Supplementary Material, Fig. S5. We have also revised lines 176-181 and equation (1) on page 7 of the original manuscript to reflect this change.

- The label for the y-axis of Figure 3 is a bit confusing because it is frequency expressed in inverse centimeters. They should either present wavenumber or report results in terms of inverse second. Similarly, “kappa” and “gamma” should be reported in inverse second.

Response: We apologize for the confusing notation used in this figure. In the revised manuscript we have converted the units of all relevant parameters to $(\text{sec})^{-1}$. This includes in Figure 3b as well as in Supplementary Figures S6-S9 (now Supplementary Figs. S10 – S13).

- Regarding the presentation, there are potential readers from mathematical physics, therefore it would be nice to better present the experimental platform. I suggest authors bringing the supplementary figure S1 to the main text and perhaps even adding an illustrative schematic of their TEM image.

Response: We thank the reviewer for the constructive suggestion on how to improve the presentation to appeal to a broader audience. We have added the TEM image to Figure 1 in the revised manuscript which also includes an SEM image of the microdisk structures and microdisk rims. A sentence describing the new Figure 1a was added to the end of the paragraph ending on line 99 on page 4 of the original manuscript. Additionally, A schematic TEM image showing the effect of DIAH cross-linking was included in the Supplementary Information as Supplementary Fig. S1.

- It would be fair to mention the following reference which is the first work suggesting (theoretically) the use of coupled-cavity PT laser for selective filtering of modes:
Miri, M. A., LiKamWa, P., & Christodoulides, D. N. (2012). Large area single-mode parity–time-symmetric laser amplifiers. *Optics Letters*, 37(5), 764-766

Response: We agree that this seminal paper deserves to be recognized in our work and have added it to the revised manuscript, as new Ref. 21. It is referenced in the revision of lines 45-47 on page 2 and of lines 164-165 on page 7 of the original manuscript.

Reviewer #2 (Remarks to the Author):

The manuscript by Evan Lafalce et al presents the spectrum purification of a defect micro-disk laser by forming exceptional points with another micro-disk laser. While the defect-introduced mode splitting between CW and CCW modes have been investigated, the results presented here are innovative towards exceptional point-related novel control on laser actions. Overall, I found the results are suitable for Nature Communications if the following comments and suggestions are well addressed:

(1) In the fabrication of the devices, the synthesis method with multiple chemicals presented in reference [22] was applied. While the fabrication approach is different, a similar coupled cavity structure using InGaAsP was described in [16] where single mode lasing was demonstrated. My question is: why using a very similar configuration, different results with multi-mode lasing were achieved (for example, Figure 2)?

Response: The reviewer brings up an interesting point about the multi-mode emission in our samples. We believe the *intra-disk* coupling induced by the defects in our microdisk pairs circumvents the mode suppression via the PT-symmetric arrangement as in the case of the work of Hodaei et al. In fact, we have observed, in cases where at least one disk is free of defects, significant suppression of higher order modes. An example is shown below. A detailed description of localization/delocalization behavior through the interplay between intra- and inter disk coupling in our multimode resonators is the goal of ongoing investigations using synthetic notch implants, as will be described in more detail in the response to Comment 3 below.

Figure R3 was added to the Supplementary Information as Supplemental Fig. S14. The discussion about the role of intra-disk coupling in our system circumventing mode suppression was added to the text in the Discussion section on page 12, immediately following the description described in the response to Comment 2 below.

Figure R3. Laser emission spectra from a coupled microdisk pair, where the pair is placed at different locations in the pump beam spot, such that only the left (Left panel) or right (Right Panel) microdisk is pumped, or the pair is pumped evenly (Center panel). The left microdisk of the pair shows significant suppression of the side modes compared to the evenly pumped pair or the right disk with substantial mode splitting due to the presence of defects.

(2) Unidirectional lasing may be expected if either CW or CCW mode is removed from the spectrum. Has this been tested? The author could comment more in-depth on the characteristics of the degenerated mode at the exceptional point.

Response: The reviewer brings up an interesting point about the influence on directionality of modes as the system passes through the EP. Unfortunately, we cannot measure the directionality within the limits of our microcavity system and experimental apparatus. Our detection schemes are based on free-space coupling and cannot resolve the directionality of the emission as in other works, for instance using microtoroid coupled to waveguides where different ports correspond to different directions of emission, as in Ref. 19 of the revised manuscript.

We note however, that the properties of EPs in three-mode systems have been discussed theoretically, notably by Demange and Graefe (Demange, J., and Graefe, E., *J. Phys. A.* **45** 025303 (2012)). In that work, they distinguish between a third-root branch point between three-modes and a square root branch point between two modes with the third mode unperturbed to leading order. The latter seems to be a more accurate description of our experimental scenario. In this case, all of these properties of chirality, mode-exchange, and WGM mode directionality that have been well-described for two-mode EPs are expected to hold. We hope to experimentally analyze the emission directionality in more detail in future work related to our synthetic notch resonators.

A brief description of the conclusions from the above reference and its implications for the directionality of modes at EP1 has been added to the manuscript at the beginning of the Discussion section beginning on line 273 on page 12 relative to the original manuscript. The reference was added as Reference 33.

(3) The manuscript should clearly present how the defect works to split the CW and CCW modes. The defect is not visualized in the SEM pictures. How was the defect designed and fabricated?

Response: As described in lines 129 – 142, with reference to Supplemental Figures 2 and 3, the defects presented in the manuscript are the natural result of the variability of the fabrication that involves building the resonators up from a solution-based method and through further fabrication steps during the removal of the photo-pattern template. The defects as visualized in the Supplemental Figs. S3 and S4 (formerly S2 and S3) are small notches in the circumference that

disrupt the boundary and lead to scattering of the laser modes, as demonstrated in the revised Figure 1e (formerly Figure 1d). We have also seen similar results in microcavities with synthetic notch implants as described below.

Figure R4. Designed defect by synthetic notch implementation. (a) Emission spectra from a disk with a small synthetic notch implant with a circumferential depth of 200nm. The large degree of mode splitting is apparent. (b) Emission spectra from a disk with a large synthetic notch implant with a circumferential depth of 800nm. The mode-splitting is entirely absent. (c) Collection-angle dependence of the emission in the plane of the microdisk with large synthetic notch as in (b) with fluorescent image at the center.

In Figure R4a, we show the spectrum from a resonator with a small synthetic notch implant implemented via incorporation of the structure in the photo-lithographic mask. The strong mode splitting in the emission spectrum is evident. We have further found that when the size of the notch is large, as in Figure R4b, the splitting is removed. Following similar work involving nano-particle scattering induced mode-splitting, we believe the removal of mode-splitting by a large notch is due to increased loss applied selectively to one of the broken-degeneracy modes. In Figure R4c, we show that the notch leads to highly directional light scattering, similar to the defects in Figure 1e of the revised manuscript (formerly Figure 1d). These notches act as scatterers in a similar manner as the uncontrolled defects in the samples described in the manuscript, which couples the CW and CCW modes via scattering into one another, thus generating a mutual dependency of their amplitudes.

Figure R5. Emission spectra from a pair of microdisks with large (800nm depth) synthetic notch. When the individual microdisks of the pair are pumped in the asymmetric pumping condition (green and red), no mode splitting is observed. When the pair is pumped evenly (blue), mode splitting is observed.

Furthermore, in coupled-pairs of microdisks with large synthetic notch implants, we observe mode-splitting when the pair is pumped evenly, and the absence of this splitting in the asymmetric regime, thus recovering the expected behavior similar to Ref. 16 of the manuscript, as shown in figure R5 above. The presence of splitting also validates the evanescent coupling of microdisk separated by a distance of 400nm. From the splitting we obtain a coupling strength of $\sim 8.0 \times 10^{11} \text{ s}^{-1}$, which is notably higher than that obtained from the FDTD simulations described above in the response to reviewer 1, and consistent with the values necessary to drive the coalescence at EP1. Moreover, these results suggest the role of intra-cavity coupling in obscuring the inter-cavity interaction in the presence of gain/loss contrast variation.

We have added Figures R4 and R5 to the Supplementary Information as Supplementary Figs S15 and S16, and discussion of these results was added to the Discussion section on page 12 immediately following the discussion described in the response to Comment 1 above.

(4) The lasing condition is determined by the threshold pumping intensity which is 29 $\mu\text{J}/\text{cm}^2$ stated in line 112. How was the threshold intensity decided? The light-light curve (the I-V curve) should be attached. This is important because we may not be able to tell the differences between the disordered spectrum of illumination under the lasing condition and the spectrum splitting.

Response: The light-light curve and the corresponding spectra at different pump intensities are shown below in figure R6. The threshold behavior in power output above $29 \mu\text{J}/\text{cm}^2$ is emphasized by the asymptotic lines showing the transition from sub-linear intensity growth in the spontaneous emission regime to the super-linear increase due to amplification by stimulated emission, which is related with the emergence of cavity modes in the emission spectrum. Figure R6 was added to the Supplemental information as Supplemental Figure S1.

Figure R6. (a) Light-light curve for an isolated microdisk. The green line indicates the region of sub-linear increase where only spontaneous emission is observed. The red line indicates the super-linear increase above threshold that corresponds with the emergence of cavity modes. (b) Emission spectra at different levels of pump fluence. The cavity modes become apparent at a fluence of $29 \mu\text{J}/\text{cm}^2$.

(5) In line 160, more convincing results should be given to the coupling between micro-disks to validate that the evanescent field can provide adequate coupling. For example, how much, with a certain spacing, is the coupling strength between mode 1 and 3, mode 2 and 3, respectively?

Response: This was clearly an important issue that needed to be addressed in our manuscript, as both Reviewers have asked about the existence of the coupling and its magnitude. Please see our response to Reviewer #1's (first and second comments), in addition to considering the experimental evidence presented in microdisk pairs with synthetic notch implants in figure R4. We note that the coupling considered in the FDTD simulations and reference above in figures R1 and R2 correspond to γ_{23} , the inter-disk coupling between modes 2 and 3, in the 3-mode model that we used in the text. As for γ_{13} , we have noted in Figure 4 of the manuscript that the splitting between the two EPs is given by $\gamma_{13}/2$. This means that to be consistent with the experiment we only need γ_{13} to be comparable or less than the linewidth of our laser modes of $\sim 3 \times 10^{11} \text{ s}^{-1}$. We believe the expected 90° phase shift between broken degeneracy WGMs that underlies the works of Zhu et al. (Refs. 32,34 in the revised manuscript; 25, 26 in the previous version) is the reason behind the asymmetry in the interdisk coupling coefficients, i.e. the maximal anti-node to anti-node overlap between modes 2 and 3 implies minimal overlap between anti-nodes of modes 1 and 3.

A brief discussion about the role of the phase shift between broken degeneracy modes was added to the end of line 270 on page 11 of the original manuscript.

(6) The introduction part reads great for presenting the concepts of non-Hermitian system and exceptional points. To make the introduction more complete, I suggest the authors including a few references on exceptional point lasers, such as Science 353, 464 (2016); PNAS 113, 6845 (2016); Opt. Lett. 39, 2423 (2014); and a recent ASAP publication in ACS Photonics DOI: 10.1021/acsp Photonics.8b00800.

Response: We thank the reviewer for the compliment and for the excellent suggestions of references that have made the introduction and references more complete and up-to-date. The four references were added as References 24-27 in the revised manuscript and cited the revision of lines 45-47 on page 2 and of lines 164-165 on page 7 of the original manuscript.

In conclusion, I think the subject itself is interesting that the spectrum splitting by defects can be turned off through exception points formed by the asymmetric coupling with a third mode.

REVIEWERS' COMMENTS:

Reviewer #1 (Remarks to the Author):

The authors have suitably addressed all my suggestions and comments. I believe the manuscript is significantly improved after this revision and I see no reason to delay its publication in Nature Communications.

Reviewer #2 (Remarks to the Author):

The manuscript by E. Lafalce et.al. has been carefully revised, including additional evidence. The work is interesting and can certainly benefit spectral purification in coupled laser systems. I would like to recommend its publication after one last minor comment is addressed:

In simulations for Figs. S6 and S7, the micro-disks sized at $D = 4 \text{ um}$ were applied, while in experiments the size of the disks was 26 um . Apparently, their coupling strengths cannot be the same. The coupling strength associated with the 4 um one should be expected stronger due to its larger waveguide bending and thus stronger evanescent wave interactions. This deviation between simulations and experiments should be further clarified in the manuscript.

REVIEWERS' COMMENTS:

Reviewer #1 (Remarks to the Author):

The authors have suitably addressed all my suggestions and comments. I believe the manuscript is significantly improved after this revision and I see no reason to delay its publication in Nature Communications.

Response: We are happy to have satisfied the reviewer's comments and thank them for the recommendation for publication 'as is'.

Reviewer #2 (Remarks to the Author):

The manuscript by E. Lafalce et.al. has been carefully revised, including additional evidence. The work is interesting and can certainly benefit spectral purification in coupled laser systems. I would like to recommend its publication after one last minor comment is addressed:

In simulations for Figs. S6 and S7, the micro-disks sized at $D = 4 \text{ um}$ were applied, while in experiments the size of the disks was 26 um . Apparently, their coupling strengths cannot be the same. The coupling strength associated with the 4 um one should be expected stronger due to its larger waveguide bending and thus stronger evanescent wave interactions. This deviation between simulations and experiments should be further clarified in the manuscript.

Response: For the simulations shown in Fig. S6 the disks diameter is $D = 26\mu\text{m}$. While we stated this in the Methods section, we did not specifically mention this in the caption of Suppl. Fig. 6. We have clarified in the caption that the simulated results shown in this figure were calculated using a realistic size disk.

For the simulations shown in Fig. S7, a smaller disk radius was used so that the mode profiles could be adequately visualized. In the caption of this figure we note that for disks of larger diameter the length of the channel between the two disks is increased, which will increase the coupling strength according to the coupled mode analysis of Little *et al.* [Little, B. E. *et al.*, Microring Resonator Channel Dropping Filters, *Journal of Lightwave Technology*, **15**, 998-1005 (1997)].

We have added this reference to support our claim, while also noting that increased bending loss, which is not accounted for in the theory of Little *et al.*, may also lead to increased coupling compared to the experimental system.